# Leveraging Interpretable Feature Representations for Advanced Differential Diagnosis in Computational Medicine

**DOI:** 10.3390/bioengineering11010029

**Published:** 2023-12-26

**Authors:** Genghong Zhao, Wen Cheng, Wei Cai, Xia Zhang, Jiren Liu

**Affiliations:** 1School of Computer Science, Engineering Northeastern University, No.195 Chuangxin Road Hunnan District, Shenyang 110169, China; 1810626@stu.neu.edu.cn; 2Neusoft Research of Intelligent Healthcare Technology, Co., Ltd., No.175-2 Chuangxin Road Hunnan District, Shenyang 110167, China; cai.wei@neusoft.com; 3The Liaoning Provincial Key Laboratory of Interdisciplinary Research on Gastrointestinal Tumor Combining Medicine with Engineering, Shenyang 110042, China; 4Department of Neurosurgery, Shengjing Hospital of China Medical University, Shenyang 110004, China; cmu071207@163.com; 5Neusoft Corporation, No.2 Xinxiu Road Hunnan District, Shenyang 110179, China

**Keywords:** differential diagnosis, interpretable feature representation, Latent Dirichlet Allocation, textual medical record, cartesian coordinate system

## Abstract

Diagnostic errors represent a critical issue in clinical diagnosis and treatment. In China, the rate of misdiagnosis in clinical diagnostics is approximately 27.8%. By comparison, in the United States, which boasts the most developed medical resources globally, the average rate of misdiagnosis is estimated to be 11.1%. It is estimated that annually, approximately 795,000 Americans die or suffer permanent disabilities due to diagnostic errors, a significant portion of which can be attributed to physicians’ failure to make accurate clinical diagnoses based on patients’ clinical presentations. Differential diagnosis, as an indispensable step in the clinical diagnostic process, plays a crucial role. Accurately excluding differential diagnoses that are similar to the patient’s clinical manifestations is key to ensuring correct diagnosis and treatment. Most current research focuses on assigning accurate diagnoses for specific diseases, but studies providing reasonable differential diagnostic assistance to physicians are scarce. This study introduces a novel solution specifically designed for this scenario, employing machine learning techniques distinct from conventional approaches. We develop a differential diagnosis recommendation computation method for clinical evidence-based medicine, based on interpretable representations and a visualized computational workflow. This method allows for the utilization of historical data in modeling and recommends differential diagnoses to be considered alongside the primary diagnosis for clinicians. This is achieved by inputting the patient’s clinical manifestations and presenting the analysis results through an intuitive visualization. It can assist less experienced doctors and those in areas with limited medical resources during the clinical diagnostic process. Researchers discuss the effective experimental results obtained from a subset of general medical records collected at Shengjing Hospital under the premise of ensuring data quality, security, and privacy. This discussion highlights the importance of addressing these issues for successful implementation of data-driven differential diagnosis recommendations in clinical practice. This study is of significant value to researchers and practitioners seeking to improve the efficiency and accuracy of differential diagnoses in clinical diagnostics using data analysis.

## 1. Introduction

Differential diagnosis [1] occupies a crucial position in the clinical diagnostic and treatment process, and it has a profound impact on ensuring medical quality, improving treatment effects, and reducing misdiagnosis rates. Differential diagnosis is a method in which clinical doctors distinguish and compare various possible diseases based on the patient’s medical history, signs, laboratory test results, and other information, ultimately determining the most likely diagnosis [2]. In this process, doctors need to possess extensive professional knowledge, keen observation skills, and rigorous logical thinking while also paying attention to various factors such as patients’ psychological needs and living environments.

Differential diagnosis is crucial in clinical diagnostics and treatment. In clinical settings, numerous diseases present with similar symptoms, complicating accurate identification. Through differential diagnosis, doctors pinpoint the most likely condition, ensuring precise, personalized treatment plans. This not only enhances treatment outcomes and reduces medical costs, but also boosts patients’ life quality and safeguards doctor–patient relationships by reducing misdiagnoses.

However, differential diagnosis is not straightforward. Many diseases show similar or atypical symptoms, requiring doctors to discern subtle differences. In the current era dominated by evidence-based medicine guiding clinical processes, there are two approaches to clinical diagnosis. Firstly, physicians must summarize the confirmed clinical presentations of the current patient and consider differential diagnoses with similar clinical manifestations. By analyzing the differences between these differential diagnoses and the patient’s actual condition, they systematically exclude other diagnoses to ultimately determine the patient’s clinical diagnosis. The second approach involves physicians initially deducing the patient’s disease diagnosis based on clinical presentations. However, for rigor, they must consider differential diagnoses with similar manifestations and then evaluate the patient’s actual clinical presentations against these to rule out the possibility of those differential diagnoses. In practice, there is a substantial likelihood that physicians need to interview patients or conduct additional laboratory tests and imaging studies to rule out differential diagnoses. Thus, the analysis and exclusion of differential diagnoses are among the most crucial steps in diagnosing a patient’s illness during clinical treatment. Comprehensive screening and exclusion of potential diseases based on the patient’s current clinical presentations and test results are key to ensuring correct treatment and pose a challenging, knowledge-intensive, and patience-demanding aspect of clinical diagnosis.

Currently, clinicians refer to official clinical guidelines and draw on years of clinical experience. However, variations in the understanding of guidelines and the depth of experience lead to differences in the capability of differential diagnosis among physicians. Less experienced doctors naturally have inferior differential diagnosis skills compared to their more experienced counterparts. The current situation has resulted in an average misdiagnosis rate of 27.8% in Chinese clinical settings [3]. In the United States, which boasts the most developed medical resources globally, approximately 12 million patients are misdiagnosed annually. It is estimated that each year, 795,000 Americans suffer death or permanent disability due to diagnostic errors. Researchers have focused on the 15 diseases causing the most harm and found that vascular events, infections, and cancers (collectively referred to as the “Big Three”) account for 75% of these severe injuries. The average misdiagnosis rate across all diseases is estimated to be 11.1%, with the rates varying from 1.5% for cardiac diseases to 62% for spinal abscesses [4].

Our survey in the industry regarding “how to assist doctors in identifying necessary differential diagnoses based on diagnosing a patient’s condition” or “how to recommend valuable differential diagnoses to doctors” reveals a surprising gap. Particularly in the field of computer science, where big data and artificial intelligence technologies are advanced, most researchers focus on using data science techniques to assist doctors in focusing on a specific disease diagnosis and excluding the patient’s possibility of other diseases. For example, assisting patients with atrophic gastritis in differentiating whether it is gastric cancer, or differential diagnosis analysis for Alzheimer’s disease patients from other cortical dementias. While these studies are invaluable in aiding accurate diagnosis, the field of assisting physicians in identifying necessary differential diagnoses for a patient remains largely unexplored.

In our research on recommending differential diagnoses to doctors, ensuring the explainability of computational processes for evidence-based medicine has become increasingly important. Presenting the computational process in an interpretable and visualized manner is vital for doctors to make critical decisions and judgments. Recent advancements in medical artificial intelligence have introduced various clinical diagnostic tools. They assist in image analysis, genetic testing, etc., to support clinical doctors. The interpretability of AI computations is crucial for patient safety in medicine. Doctors need transparent AI operations to ensure reliability and patient safety. Black-box computations are inappropriate, as medical decision-making involves multiple variables. Understanding the principles behind AI’s diagnoses is crucial for ensuring optimal treatment.

The main contributions of this study are as follows:We developed a method leveraging extensive clinical data to assist clinicians in recommending differential diagnoses that require discernment. This clinical necessity has historically remained inadequately addressed.Our proposed machine learning framework is fully processable through interpretable representations and visualizations, allowing doctors understanding the reasons and processes behind the recommended differential diagnoses, achieving full-process white-box computation for traceability in evidence-based medicine scenarios.In our framework, we use the LDA [5] topic model. Facing the challenge of determining reasonable topic numbers without prior knowledge, we propose an interpretable representation method to assess the quality of topic models and optimize topic numbers, achieving precise optimization of model quality.

The rest of the paper is arranged as follows. Section 2 introduces the latest developments in the research on interpretable representation of medical text data and the use of medical artificial intelligence technology to assist doctors in clinical diagnosis. Section 3 introduces the detailed calculation process for extracting interpretable representations from medical text data and using past case data for differential diagnosis decision support. Section 4 provides a detailed introduction to the experimental process and results using real clinical data. Section 5 summarizes the current work and looks forward to future research.

## 2. Related Work

The utilization of data science for assisting physicians in computationally identifying differential diagnoses necessary for patients is immensely beneficial. However, upon reviewing an extensive array of research papers, it is observed that a substantial number of studies primarily focus on assisting medical professionals in the diagnosis and exclusion of specific or classes of diseases within particular disease subtypes. Undoubtedly, this type of research is invaluable as it aids in enhancing diagnostic accuracy in specialized medical fields. Nonetheless, research is less common in the area of computational assistance for physicians to accurately identify alternative diagnoses for existing clinical presentations and preliminary diagnoses made by the doctors. Most of the relevant content is discussed as part of Clinical Decision Support Systems (CDSS) [6] or Disease-Specific Decision Support Systems, yet there is a dearth of in-depth studies specifically targeting this aspect. We reviewed research outputs related to our proposed study question from the past three years and summarized several findings that partially overlap with the content of our study.

The research by P. Guimarães and colleagues [7] is a pioneering endeavor in the application of artificial intelligence for gastrointestinal health. Their work intricately combines endoscopic imagery with clinical data to differentiate between various forms of colitis, including Inflammatory Bowel Disease (IBD), infectious, and ischemic types. This study is notable for its innovative use of a Convolutional Neural Network (CNN), trained on a substantial dataset of 1796 images from 494 patients. In parallel, the study explores a Gradient Boosted Decision Trees (GBDT) algorithm, leveraging five critical clinical parameters. A unique aspect of this research is the exploration of a hybrid approach, merging CNN and GBDT methodologies. The findings reveal intriguing variations in accuracy and predictive values across the CNN, GBDT, and hybrid models. A key insight from this study is the superior global accuracy of the GBDT algorithm, which relies on clinical parameters, over both the CNN and expert image classification. The research concludes with a candid acknowledgment of the limitations of AI systems solely based on endoscopic image analysis and underscores the potential for enhanced performance with more diverse image datasets in the future.

The study by D. Cha and team [8] undertakes a comparative analysis of deep learning models versus human expertise in diagnosing ear diseases using otoendoscopic images. This research is distinguished by its extensive use of a dataset comprising 7500 images across six disease categories. It meticulously evaluates the performance of deep learning (DL) models against that of clinicians. A striking finding is the high accuracy of DL models, though they exhibited a tendency for bias toward more prevalent diseases. In contrast, human physicians, despite being less accurate overall, showed less bias toward disease prevalence and demonstrated significant variability in their diagnoses. The study culminates with the suggestion that a synergistic approach, combining the strengths of DL models and human expertise, can significantly enhance patient care, particularly in regions with limited access to specialist care.

Y. Miyachi and collaborators [9] introduce an innovative clinical decision support system (CDSS) aimed at revolutionizing differential diagnosis in medical practice. This system, utilizing machine learning with a learning-to-rank algorithm, stands out for its remarkable performance, surpassing conventional models in predicting potential diseases. It addresses a critical challenge in medical practice: diagnostic errors. By integrating a comprehensive database of confirmed and differential diseases, the system aligns closely with real-world diagnostic processes. Its strength is particularly evident in handling complex cases and rare diseases, providing a robust tool for medical decision-making and mitigating reliance on physician intuition to avoid confirmation bias.

The work by C. Brown and associates [10] delves into the integration of AI tools in clinical settings, with a focus on a case study involving delayed diagnosis in a 27-year-old woman. This paper sheds light on how AI can aid in mitigating cognitive biases and errors in the diagnostic process. It underscores AI’s potential in enhancing diagnostic accuracy and addressing human factors in medical errors, such as augmenting real-time medical imaging interpretation and aiding in differential diagnosis. The study also highlights the risks associated with AI, such as automation bias and data quality issues, and emphasizes the importance of clinician training in AI interpretation. The paper underscores the crucial need for clinicians to develop skills for responsible AI adoption in healthcare and advocates for patient safety in the utilization of health data.

Y. Harada and colleagues [11] investigate the influence of an AI-driven Diagnostic Decision Support System (DDSS) on physicians’ diagnostic decisions. In a randomized controlled trial involving 22 physicians and 16 clinical vignettes, the study evaluates the impact of AI-generated differential diagnosis lists. The findings reveal a significant influence of the AI list on physicians’ diagnoses, with at least 15% of differential diagnoses being impacted. The study shows a higher prevalence of diagnoses identical to the AI’s suggestions in the group receiving AI assistance, illustrating AI’s potential in enhancing diagnostic processes. However, the varied level of trust in AI among physicians is a noteworthy aspect, indicating the complexity of integrating AI into medical decision making.

## 3. Methods

This chapter delves into the use of interpretable data representation in aiding differential diagnosis recommendations, premised on clinical text data. It begins by outlining the clinical context in which physicians engage in diagnostic and differential analysis, emphasizing the systematic and multidisciplinary approach adopted, starting with an in-depth patient evaluation—from history taking to physical exams and advanced diagnostic tests. This rigorous process informs the preliminary diagnostic hypotheses. Subsequently, a differential diagnosis is performed to refine these hypotheses, employing specialized tests if necessary, leading to a definitive diagnosis and therapeutic strategy, all documented within the patient’s medical records.

Central to this research is leveraging the wealth of information within clinical medical records, enhancing data-driven decision making in the differential diagnosis phase. The chapter introduces a novel technique for extracting interpretable data representations from medical texts, promising to augment this critical aspect of clinical diagnostics.

## 4. Interpretable Textual Medical Record Representation

In natural language processing, the BERT model [12] has emerged as a prominent figure in text representation, overshadowing traditional methods like the Latent Dirichlet Allocation (LDA) topic model in various applications. Nonetheless, LDA maintains a competitive edge in certain domains, particularly due to its superior interpretability. As a generative model utilizing bag-of-words [13] and Dirichlet distribution [14], LDA interprets text data by identifying multiple topics and their pertinent keywords, offering a clear, intuitive insight into textual structure and semantics. This contrasts with BERT’s pre-trained mechanisms, which, despite their performance benefits, encapsulate complexities that challenge straightforward human understanding. In contexts demanding high interpretability, such as clinical applications, LDA’s transparency signifies its ongoing relevance.

In natural language processing, the BERT model excels in text representation. However, in complex applications, the traditional LDA topic model maintains competitiveness, particularly in interpretability. LDA, a generative model, distinguishes itself by intuitively depicting text structures and semantics, simplifying human interpretation. Despite BERT’s high performance, its less transparent internals pose challenges. In interpretability-sensitive areas like clinical scenarios, LDA’s clarity is indispensable.

Additionally, LDA’s computational efficiency is crucial under resource constraints. BERT, though powerful, requires extensive resources, posing issues for real-time, hardware-limited analysis. LDA’s lower demands make it suitable for such situations, particularly in real-time medical contexts.

Moreover, LDA aligns with medical differential diagnosis, effectively paralleling the extraction of patient symptom information to identify possible ailments. This natural alignment underscores LDA’s relevance in capturing patient–disease correlations.

While other methods like TF-IDF and word2vec [15] exist, they fall short on certain aspects. TF-IDF lacks thematic insights, and word2vec, though strong in semantic relations, does not grasp overarching text themes as LDA does. Therefore, for medical differential diagnosis, LDA remains the superior option due to its interpretability, efficiency, and thematic representation capabilities.

On the other hand, in the mentioned scenario of assisting differential diagnosis calculations, the ultimate task is to find parallel diagnoses similar to the current patient for doctors’ reference by calculating the similarity between the patient’s condition information and the known diagnosis and condition features mapped in accumulated medical records. From the perspective of similarity calculation, the LDA topic model has a significant advantage. The LDA topic model is a generative probabilistic model based on the bag-of-words model and Dirichlet distribution, which can represent text as a distribution of multiple topics. This method can intuitively present the structure and semantic information of text data. When calculating similarity, the similarity can be assessed by comparing the distance between document topic distributions (such as KL divergence [16], Jensen–Shannon divergence). This method is relatively simple and intuitive in the calculation process, and due to the sparsity of topic distributions, the calculation efficiency is higher. Although the BERT model, as a pre-trained deep learning model, has a significant advantage in performance, its internal representation is relatively difficult to directly use for similarity calculation [17]. First, if the BERT model itself is not trained simultaneously with downstream tasks, part of the BERT model’s advantages will be wasted. The BERT model uses a deep bidirectional encoder of Transformer [18], capturing the polysemy of words in different contexts through dynamic context modeling. However, using BERT only as word embedding greatly discounts this rich dynamic semantic representation capability. Extracting BERT’s output as static word embeddings means that we only focus on the representation of individual words, ignoring the context information, which cannot fully exploit BERT’s advantages in higher-level semantic modeling. Moreover, BERT transfers pre-trained knowledge to downstream tasks through a fine-tuning process, achieving significant performance improvements in various specific tasks [19]. In the process of modeling using the BERT model, it is customary to utilize text data labeled for downstream tasks for training. This approach enables the conversion of textual data into representations that are more characteristic of the features pertinent to downstream tasks. However, if we only use BERT as word embedding, this process will be ignored, causing BERT to be unable to fully adapt to downstream tasks. In this case, the representation based on BERT is not optimized for specific tasks, weakening its performance in the task.

Therefore, choosing to use the Latent Dirichlet Allocation model for interpretable representation of medical text data in this study is a very competitive choice. The detailed process of constructing the interpretable representation of text medical records using the Latent Dirichlet Allocation model is as follows.

First is the selection of medical record data. Currently, the management of patients’ text-based medical records in large hospitals in China is quite similar [20]. For inpatients, the content related to the patient’s illness and treatment from onset to admission is recorded in the “Course Record—First Course Record” medical record. In this process, the patient’s attending physician needs to make the initial diagnosis after admission based on the patient’s condition. While determining the diagnosis, it is also necessary to record the differential diagnosis and the reasons for the undiagnosed diagnoses in detail. These reasons are also documented in the main text of the “Course Record—First Course Record”. Therefore, the “Course Record—First Course Record” is used as the data source for constructing differential diagnosis assistance prompts. In the following text, “medical record” replaces “Course Record—First Course Record”.

Next, the medical record text data are anonymized by removing the patient’s personal information. Then, Chinese word segmentation is used to split the words in the medical record text and perform part-of-speech tagging for each word. Since the core content of patient condition records in medical records consists of nouns, verbs, and adjectives, and to ensure the initial topic model’s information density by excluding long-tail words with little information value during topic model training, data representation is defined as follows:(1)$$D=\left(d_{1},d_{2},d_{3}\ldots d_{n}\right),$$
(2)$$d=\left(w_{1},w_{2},w_{3}\ldots w_{m}\right),$$
(3)$$V=\left({w^{\prime }}_{1},{w^{\prime }}_{2},{w^{\prime }}_{3}\ldots {w^{\prime }}_{v}\right),$$
(4)$$T=\left(t_{1},t_{2},t_{3}\ldots t_{k}\right).$$

*D* represents the collected set of medical record texts, *n* represents the size of the document set. *d* represents a single medical record text, *w* represents words in the medical record, and *m* represents the number of words in a single medical record text. *V* represents the corpus of words in all medical record texts, *v* represents the size of the corpus. *T* represents the set of topics under this corpus, *t* represents a single topic, and *k* represents the number of topics.

Two matrices are obtained in the LDA topic model using Gibbs sampling [21], which are *P*(*w|t*), representing the distribution probability of words within a topic, and *P*(*t|d*), representing the topic distribution probability of each document.

Since the topic model is trained under a fixed corpus, the calculation results of document–topic distribution are limited to the current corpus. To calculate the topic distribution of a new document, the following formula needs to be used:(5)$$P\left(t|d_{new}\right)=\sum_{w_{i}\in d_{new}}P\left(t|w_{i}\right)$$

In this case, $d_{new}$ represents a new document outside the training topic model. In this calculation process, $P(t|w_{i})$ is introduced as a new variable. Since *P*(*w|t*) is obtained during the training of the topic model, it can be known through the Bayesian formula:(6)$$P\left(t|w\right)=\frac{P(w|t)P(t)}{P(w)}$$

In the formula, *P*(*w*) can be understood as the word frequency in the corpus of the training topic model, which can be calculated through the following formula:(7)$$P\left(w\right)=\frac{\left|{w^{\prime }}_{i}\right|}{\sum_{{w^{\prime }}_{i}\in V}\left|{w^{\prime }}_{i}\right|}$$

In the formula, *P*(*t*) can be understood as the absolute distribution of topics in the corpus of the training topic model. Since the training data are a set of medical record texts, under the premise that the collection of medical record data covers the range of all departments and diseases in the hospital, *P*(*t*) can be considered as the probability of the occurrence of various topics in all medical record texts. Combined with the Bayesian formula [22], it can be calculated through the following formula:(8)$$P\left(t\right)=\sum_{d\in D}P(t|d)P(d)$$

In the above formula, *P*(*t|d*) is obtained by training the topic model. *P*(*d*) can be understood as the probability of any document occurring. Each medical record comes from a remote patient and is not affected by the value of any other random variable, that is, the medical records are mutually independent. At the same time, each medical record follows the same probability distribution, that is, they have the same probability density function or probability mass function, so it can be judged that the probability of the medical record itself satisfies the definition of independent and identically distributed. For the convenience of calculation, the value of *P*(*d*) is set to one. The final calculation of *P*(*t|w*) can be expressed as
(9)$$P\left(t|w\right)=\frac{P(w|t)\sum_{{w^{\prime }}_{i}\in V}\left|{w^{\prime }}_{i}\right|\sum_{d\in D}P(t|d)}{\left|w\right|}$$

For a new document $d_{new}$ outside the training topic model corpus, the topic distribution of the document can be quickly calculated through *P*(*t|w*):(10)$$P(t|d_{new})=\sum_{w_{i}\in d_{new}}P\left(t|w_{i}\right)$$

Since the words in each topic can express a part of interpretability to some extent through the distribution of words in the topic, the form of expressing the connotation of medical records through interpretability representation is achieved through the topic distribution in the medical records.

## 5. Optimal Topic Number Calculation Methods for Interpretable Topic Models

Choosing an inappropriate number of topics can have adverse effects on the performance and results of LDA topic models. This is manifested in several aspects: first, poor topic interpretability. Too few topics may make it difficult to distinguish between them, making the generated topics difficult to interpret, while too many topics may split a single topic into multiple similar subtopics, also reducing interpretability. Second, overfitting and underfitting issues. Too many topics may lead to overfitting of the model, resulting in excellent performance on the training data but poor generalization performance on new data; too few topics may lead to underfitting, resulting in poor performance on the training data, affecting the prediction ability on new data. Next, incomplete topic coverage. Too few topics may fail to capture all important topics, thus reducing the effectiveness of the model in practical applications. In addition, model complexity issues. Too many topics may make the model overly complex, increasing computational costs and storage requirements; at the same time, too many topics may also make the analysis and interpretation of results more difficult. Finally, result stability issues. An unsuitable number of topics may lead to poor model stability, with different initializations and random seeds potentially producing different results. To address these issues, it is crucial to choose an appropriate number of topics. Various methods can be tried to determine the optimal number of topics and adjusted according to specific application scenarios and requirements. The following are some algorithms for determining the optimal number of topics for LDA topic models:

Perplexity [23]: Perplexity is a metric for measuring the predictive ability of a model on a test dataset. Generally, a lower perplexity value indicates a better model. However, perplexity may not be a very intuitive indicator for choosing the optimal number of topics, as overfitting can lead to lower perplexity in some cases.

Topic Coherence [24]: Topic coherence measures the semantic similarity between words within the same topic. A higher topic coherence score usually indicates a better model. Common coherence measures include C_v, C_umass, PMI, etc. However, calculating coherence may require significant computational resources, and its applicability may vary between different domains and datasets.

Hierarchical Dirichlet Process (HDP) [25]: HDP is a nonparametric Bayesian method that can automatically learn the number of topics. However, HDP may require longer computation time and may produce more topics in some cases, leading to lower differentiation between topics.

The above methods all provide mathematical approaches for calculating the optimal number of topics. However, each algorithm has its own limitations and advantages. After analyzing the core ideas of calculating the optimal number of topics using the above methods, this study proposes the following three core ideas for the optimal number of topics:
The number of topics should be as large as possible within an allowable range; too few topics definitely cannot express enough interpretable meaning.Each topic should express an independent theme as much as possible. This can be understood as follows: if each topic is mapped to a semantic space, the larger the distance between topics, the better; situations where two topics are very close (similar) should be avoided as much as possible.The words expressing semantics within a topic should have semantics as independent as possible. This can be understood as having as few synonyms as possible. Due to the randomness in the training process of topic models, some words may have the same semantic representation in topic expression, which may potentially share the representation ability of a certain semantic dimension within the topic.


Based on the above core ideas, and combined with some intermediate results generated in the calculation process of the previous section, this study proposes a simple calculation method with easy-to-understand calculation principles and excellent practical results, called “Balance Similarity”.
(11)$$SimiTopic=\sum_{w_{i},w_{j}\in V}similarity\left(P(w_{i}|t),P(w_{j}|t)\right),$$
(12)$$SimiWord=\sum_{t_{i},t_{j}\in T}similarity\left(P(t_{i}|w),P(t_{j}|w)\right),$$
(13)$$\underset{K}{\mathrm{argmin}}BalanceSimilarity\left(K\right)=\frac{SimiTopic\times SimiWord}{K^{2}}.$$

In calculating the similarity in the topic, *P*(*w|t*) is obtained through the results of topic model training. Any two topics are represented by the probability distribution under the same vocabulary, and the similarity is calculated using methods such as cosine similarity or KL divergence. The quantified result of the similarity between topics within the model under the current number of topics is obtained through accumulation and summation.

In calculating the similarity in SimiWord, *P*(*t|w*) is obtained through the process of calculating the interpretability representation of medical record texts in the previous section. By using topics and the Bayesian formula transformation, each word is represented in the semantic space in the topic dimension. The similarity is also calculated using methods such as cosine similarity or KL divergence, and the quantified result of the similarity between words in the corpus within the model under the current number of topics is obtained through accumulation and summation.

Finally, in the denominator, the square of *K* restricts the excessive shrinkage or expansion of the semantic space when the number of topics is too large or too small, ultimately completing the evaluation of the number of topics and the quality of the topic model.

The introduction of the Balance Similarity algorithm not only offers a validation method for the optimal number of topics in topic modeling, but also distinguishes itself from mathematical optimization algorithms like Perplexity. Within the domain of topic modeling, it deeply deconstructs the “document–topic” and “topic–word” paradigms, tailoring an algorithm specifically designed to determine the optimal number of topics.

## 6. Discriminative Diagnosis Recommendation Calculation Based on Interpretable Representations

Before calculating the differential diagnosis recommendation, a computation knowledge base for differential diagnosis recommendation needs to be built based on the trained topic model. First, the text medical records are classified according to the patient’s primary diagnosis, the medical record texts under each diagnostic category are integrated into a large text, and the topic distribution of the current diagnosis is calculated through *P*(*t*|*w*).
(14)$$CD=\left({cd}_{1},{cd}_{2},{cd}_{3}...{cd}_{l}\right),$$
(15)$$D_{{cd}_{i}}=\left(d_{1},d_{2},d_{3}...{cd}_{n^{\prime }}\right),$$
(16)$$P(t|{cd}_{i})=\sum_{d_{j}\in D_{{cd}_{i}}}\sum_{w_{k}\in d_{j}}P\left(t|w_{k}\right).$$

Here, *CD* represents the clinical diagnosis set, *cd* represents a specific diagnosis, and $D_{cd}$ represents the medical record set within a diagnosis. The topic distribution of a diagnosis ${cd}_{i}$ is obtained through $P(t|{cd}_{i})$ calculation.

After the clinician records the medical history for a newly admitted patient, the topic distribution $P(t|d_{new})$ of the current medical record can be obtained using the methods in the interpretability characterization section. Using the aforementioned topic–diagnosis distribution, the following calculation method is proposed.

In Figure 1, a two-dimensional Cartesian coordinate system is used to represent the relationship between the current medical record and a clinical diagnosis. The *x*-axis represents the medical record, and the *y*-axis represents the clinical diagnosis. The diagonal serves as a line segment representing the median distance between the current clinical case and differential diagnoses. Topics (points) closer to this line are indicative of themes more closely aligned with the disease diagnosis and differential diagnoses of the current case. All topics in the topic model are represented in the coordinate system through ($P(t|d_{new})$,$\;P(t|{cd}_{i})$) two-dimensional coordinates, and the following can be judged through the positions of the topics in the coordinate system:
The closer the topic coordinate is to the *x*-axis, the more representative the topic is in the medical record compared to the current clinical diagnosis. If the topic has a larger weight in the topic distribution of the medical record, it can be used as a key interpretative topic to prove that the current diagnosis is not the differential diagnosis of the medical record.The closer the topic coordinate is to the *y*-axis, the more representative the topic is in the clinical diagnosis compared to the current medical record. If the topic has a larger weight in the topic distribution of the medical record, it can be used as a key interpretative topic to prove that the current diagnosis is not the differential diagnosis of the medical record.The closer the topic is to the bisector of the coordinate system, the more it has equal proof reference value for the current diagnosis and medical record, and it may be a very important basis for judging the possibility of differential diagnosis.The farther the topic is from the origin, the more important the feature is in calculating the differential diagnosis recommendation.


Based on the above core ideas, the following calculation process is constructed:(17)$${distance}_{t_{i}}=\sqrt{{P(t_{i}|d_{new})}^{2}+{P(t_{i}|cd)}^{2}},$$
(18)$$\theta_{i}={tan}^{-1}\frac{P(t_{i}|d_{new})}{P(t_{i}|cd)}.$$

The angle range is set to θ′–θ″ to distinguish the bonus or penalty factors for differential diagnosis. According to the experimental results and experience judgment, it is more appropriate to set the angle between 35 and 65 degrees. In some rare disease scenarios, the angle can be relaxed to 15–75 degrees. As shown in the figure above, topics in the light blue area, falling within the θ′–θ″ range, are used as bonus factors for differential diagnosis, and topics in the dark blue area, falling outside the θ′–θ″ range, are used as penalty factors for differential diagnosis. The calculation method is as follows:(19)$${socre}_{t_{i}}=\left\{\begin{array}{c} {distance}_{t_{i}}\times \sin2\theta_{i},\;\;\theta \prime \prime >\theta_{i}>\theta \prime \\ -{distance}_{t_{i}}\times \cos4\theta_{i},\;\;\theta \prime \prime <\theta_{i}\;\mathrm{or}{\;\theta }_{i}<\theta \prime \end{array}\right.$$
(20)$$Score=\sum_{t_{i}\in T}{socre}_{t_{i}}.$$

## 7. Experiment

To substantiate the computational efficacy of the methodology delineated in the antecedent chapter for tangible clinical differential diagnosis recommendation contexts, we arbitrarily culled approximately 18,000 “medical record—progress notes—initial progress notes” (hereinafter denoted as “medical records”) from the conglomerate of initial admissions across all departments between March 2017 and April 2018 at Shengjing Hospital Affiliated to China Medical University, extracted from the Electronic Medical Records (EMR) system. The lexicon span of the corpus data extended to 31,501 entries.

### 7.1. Experiment 1: Optimal Solution Experiment for the Number of Topics in Topic Modeling

In order to ascertain the efficacy of the Balanced Similarity approach in evaluating the performance of topic models, we performed an empirical investigation utilizing the clinical medical records of 15,972 patients.

The training process used the LDA training method in the Gensim [26] framework. Perplexity, coherence, and HDP were used as model evaluation methods for the trained model. In the coherence evaluation, u_mass, c_uc, and c_npmi were used for the evaluation calculation of three parameters. The experimental results are described in Figure 2.

The figure above depicts the quantified outcomes of various evaluation metrics across different numbers of topics through visualization. The abscissa and ordinate represent the number of topics and the values of evaluation metrics, respectively. The points in the figure signify the results of evaluation metrics at various topic quantities.

The quantitative metric results of the proposed method in actual practice, compared with other Baselines under various topic numbers, are as follows in Table 1.

From the above results, it can be seen that all model evaluation methods fluctuated around the number of topics = 100. The number of topics modeled by HDP was finally 150. For other evaluation methods, their inflection points were all within the fluctuation range of around 100, and the optimal solution of the model could not be well judged from the inflection points of the data itself. The inflection point of the Balance Similarity proposed in this paper appeared the most obvious at the number of topics = 105. Therefore, this method can more accurately determine the number of topics with different parameters.

### 7.2. Experiment 2: Efficacy Assessment of Differential Diagnosis Computation

The aforementioned data and models serve as the training set. Within the collection of 15,972 medical records in the training set, a total of 1718 clinical diagnoses and 4146 differential diagnoses articulated by clinicians are encompassed. Subsequently, we solicited an additional 1500 patient records from the hospital to construct a test set, ensuring that the clinical and differential diagnoses of these patients fell within the purview of the conditions represented in the training set. Within these 1500 patient records, a total of 3645 differential diagnoses were identified, serving as comparative benchmarks for validation testing.

Currently, when physicians engage in the differential diagnosis of a patient’s clinical presentation, they generally rely on official guidelines or clinical diagnostic pathways issued by medical regulatory and guidance bodies. These sources identify the differential diagnoses to be considered for specific diseases in clinical diagnostics. Alternatively, physicians may draw upon their own experience accumulated through practical clinical work. This process involves clinicians manually entering the differential diagnoses into the clinical treatment system after careful consideration. Therefore, in our experiments, we compute and compare the differential diagnoses that clinicians arrive at through manual cognitive processes during actual clinical treatment.

The experimental procedure involved employing the proposed method to recommend a ranked ordering of differential diagnoses based on the medical record text data. The outcomes were compared with the differential diagnoses recorded by actual clinicians in the medical records, utilizing the model results from the aforementioned experiment with a thematic count of 105. On average, each medical record presented 2.43 differential diagnoses, with the record encompassing the most differential diagnoses containing 5. Hence, the selection of top N was planned by calculating the propensity scores for each medical record against all diagnoses using the method proposed in Section 3. Through ranking, the top 5 and top 10 differential diagnoses most associated with each medical record were contrasted with the actual clinical differential diagnoses (if a clinical diagnosis was hit, the subsequent diagnosis was considered in the computation). The experimental results are illustrated in Table 2.

The aforementioned experimental results reveal that, within the final experimentation, the top 5 outcomes essentially encapsulate the majority of the differential diagnoses posited by clinicians, while the top 10 outcomes almost entirely encompass the differential diagnoses determined by the clinicians. Hence, it can be ascertained that the current experimental method is efficacious within a clinical context.

For the differential diagnoses, a two-dimensional coordinate system can be utilized to provide clinicians with an interpretable assessment. Below is an illustrative example in the form of a Cartesian coordinate system that differentiates between “Acute Appendicitis” and “Acute Cholecystitis” for a patient diagnosed with the former (the illustration omits long-tail topics).

In Figure 3, the scenario is illustrated where a patient with acute appendicitis is involved in the differential diagnosis process in comparison to acute cholecystitis. Each point in the figure represents a topic extracted from the current patient’s medical record, positioned on the graph based on its relevance in diagnosing the two diseases. It is evident that although the patient was diagnosed with acute appendicitis by the physician, the actual clinical manifestations closely resemble those of acute cholecystitis. In this context, it is crucial for the clinician to discern whether the patient might be at risk of acute cholecystitis. In the actual clinical diagnosis and treatment process, clinicians need to scrutinize the ultrasonography report, particularly for the presence of gallstones at the gallbladder site, potentially leading to acute inflammation. This differential analysis process should be documented in the patient’s medical record as per clinical guidelines. The methodology proposed in this study can aid clinicians in swiftly identifying the necessary differential diagnoses for the patient.

Example of a topic model:

The yellow node represents the following topics: (Abdominal Pain: 0.25213, Upper Abdomen: 0.15843, Nausea: 0.03581, Paroxysmal: 0.03412, Anorexia: 0.03178, Constipation: 0.02631, CT Scan: 0.02123, Effusion: 0.02015, Swelling: 0.01561, Digestive System: 0.01321, Routine Blood Test: 0.01164, Tenderness: 0.00975, Bowel Sounds: 0.00716, Pain Transfer: 0.00654, Surgical Procedure: 0.00621, Perforation: 0.00431, Resection: 0.00411, Medication Treatment: 0.00349, Colic: 0.00184, Traditional Chinese Medicine: 0.00094).

The red node represents the following topics: (Fever: 0.19834, Chills: 0.18426, Severe Pain: 0.17534, Muscle Rigidity: 0.09642, Shoulder Area: 0.09043, Indigestion: 0.07642, Belching: 0.07234, Healing: 0.07049, Shock: 0.04824, Intestinal Obstruction: 0.04754, Antibiotics: 0.04359, Gallbladder: 0.02047, Emergency: 0.02001, Tenderness on Palpation: 0.01045, Peritonitis: 0.01008, Suppuration: 0.00843, Cancer: 0.00724, Greasiness: 0.00702, Digestive System: 0.00497, Pain: 0.00467).

In conclusion, the precision of the applied results of the topic model training was evaluated for the differential diagnosis recommendation by assessing Balance Similarity. Utilizing the optimal model with a topic count of 90, determined through perplexity calculation, and the model with an optimal topic count of 150, ascertained via Hierarchical Dirichlet Process, computations for differential diagnosis recommendations were performed. The evaluation was based on the indicators of top 5 and top 10 ranges, with precision calculations described in Table 3.

The results elucidated that the outcomes derived from the Balance Similarity calculation were superior.

Given the scarcity of research pertaining to differential diagnosis recommendations, there is a dearth of suitable alternative methods for baseline comparison. To substantiate the efficacy of the proposed method in utilizing topic models for interpretable representation calculations, the experimental process was designed to include the following procedure for baseline comparison.

The assessment of whether a diagnosis could serve as a differential diagnosis for a medical record was performed by calculating the similarity between the topic distribution of the medical record and that of the topic distribution of the collective medical records for each diagnosis. The evaluation was again based on the indicators of top 5 and top 10 ranges, with results described in Table 4.

The aforementioned results indicate that the outcomes derived from the similarity calculations using the topic distribution in the topic model exhibit significant deviations compared to the proposed method and the diagnoses by clinicians. This serves as evidence of the efficacy of the method proposed in this study for the computation of interpretable representation in differential diagnosis recommendations using topic models.

To validate the statistical difference between the training and test sets, a significant difference analysis was performed using the basic information of 15,972 training case sets and 1500 verification case sets. The detailed results are presented in Table 5.

In the realm of medical data analysis, the heterogeneity of patient populations within training and testing datasets can introduce significant variability, particularly when the datasets exhibit disparate disease symptomatology. This divergence precludes the use of certain clinical data metrics, such as laboratory test results, for the calculation and comparison of significant differences due to their intrinsic dependency on the disease state. Therefore, to circumvent this limitation, the present study employed demographic fundamentals, including gender, age, marital status, blood type, and Rh factor, as proxies for conducting a comparative analysis of the differences.

The selection of these demographic parameters was grounded in their relative stability and independence from the patients’ clinical conditions, thereby providing a neutral baseline for comparative evaluation. The statistical analysis was executed using appropriate tests to ascertain the presence of significant differences between the training and testing cohorts.

As illustrated in the preceding Table 5, the statistical analysis revealed no significant differences between the patients encompassed within the training and testing datasets involved in the differential diagnosis experiment (*p* > 0.05). This absence of significant demographic disparities lends credence to the methodological soundness of the experiment and substantiates the external validity of the proposed approach. Consequently, the findings affirm the efficacy of the proposed method, underscoring its potential applicability across diverse patient populations without the confounding influence of demographic variability.

In addition to the aforementioned analysis experiments with significant differences, to verify the robustness of this method across different demographic groups, we performed cross-group testing of patient data based on age, gender, and diagnosis. It is important to note that, in accordance with international convention, age groups are divided into children and adolescents (under 18 years), middle-aged adults (19–65 years), and elderly adults (66 years and older). This classification is used to evaluate the computational efficacy and robustness of the observed significant differences across different age groups.

The experimental results in Table 6 indicate that the proposed method remains effective across different populations, without significant differences arising due to divisions by age and gender. These findings substantiate the robustness of the proposed method. Furthermore, the top 5 differential diagnosis recommendations exhibit no noticeable disparities, thereby demonstrating the consistent computational efficacy of the proposed method across diverse demographic groups.

The computations in this study were conducted using proprietary data from Shengjing Hospital of China Medical University. The entire experimental process was confined within the hospital premises to ensure data security. Efforts were made to seek publicly available datasets for more open experimental testing. However, medical textual records, which include detailed descriptions of patients’ conditions and some personal information, are considered highly confidential personal data by doctors, patients, hospitals, and regulatory authorities alike. We explored renowned medical research databases such as MIMIC-III and the SEER database, which contain structured data like patient diagnosis, medical orders, and surgical procedures. However, these databases lack descriptive textual information about patients. Our proposed method utilizes patient textual records and is based on a topic model framework, rendering structured data unsuitable for our approach.

## 8. Conclusions

Differential diagnosis plays a pivotal role in the clinical diagnostic and treatment process. It is crucial in ensuring that patients are accurately diagnosed and receive the appropriate subsequent treatment. This study successfully proposed and validated an interpretable representation method based on Latent Dirichlet Allocation (LDA) topic modeling for clinical differential diagnosis recommendations. This method maps cases and relevant medical literature to low-dimensional topic spaces, enabling simplification and efficient processing of complex medical data. Our approach not only demonstrates high accuracy and recall, but also provides doctors with intuitive, interpretable association information, contributing to the improvement of clinical decision-making quality and efficiency.

Experiments showed that the interpretable representation method based on LDA topic modeling performs superiorly in evaluation metrics, showing significant competitive advantages compared to calculations based on topic distribution, and also demonstrates its core value in providing diagnostic reference for clinical doctors during the clinical diagnosis process. In addition, we also demonstrated the advantages of this method in mining associations between diseases and topic representations and providing interpretable information, thereby helping doctors gain a deeper understanding of the recommended results.

In a manner akin to the research introductions found in associative studies, the majority of diagnostic computing is currently integrated into Clinical Decision Support Systems. During the clinical diagnostic and treatment process, physicians utilize these systems to input detailed patient conditions and treatment modalities. Concurrently, the CDSS analyzes the patient’s clinical data in real time and provides relevant treatment recommendations, including differential diagnoses. However, the recommendations for differential diagnoses in such systems are often based on clinical treatment pathways or medical knowledge bases like BMJ [27] and Elsevier [28]. While recommendations based on medical knowledge bases align well with the principles of evidence-based medicine, they might not always yield accurate results due to the unique conditions of each patient. The method proposed in this article can replace the differential diagnosis prompts in the CDSS. However, a prerequisite for this is that the hospital’s clinical standards and data accumulation are at the forefront. Through in-depth discussions with physicians, it was further suggested that the computational method mentioned in this article should be modeled using data from leading hospitals and subsequently applied in lower-tier hospitals. This data and model-sharing approach aim to address the disparities in medical resources and clinical standards.

Despite the significant achievements of our research, there are some limitations and directions worth exploring. First, due to the randomness of the LDA topic model [29], the analysis results may be affected by the initial parameter settings. Future research can try to adopt more stable topic modeling methods to improve the consistency of results. Furthermore, this approach mainly focuses on text information processing and does not fully utilize various medical data such as images and genes [30]. Future research may explore interpretable representation methods that incorporate multi-modal data to further enhance the performance of clinical differential diagnosis recommendation systems.

In summary, this study successfully proposed an interpretable representation method based on the LDA topic model and achieved remarkable results in clinical differential diagnosis recommendations. We believe that with continuous optimization and improvement of the method, it will provide doctors with more accurate and interpretable diagnostic suggestions, taking a crucial step towards realizing intelligent applications in the medical field.

## Figures and Tables

**Figure 1 bioengineering-11-00029-f001:**
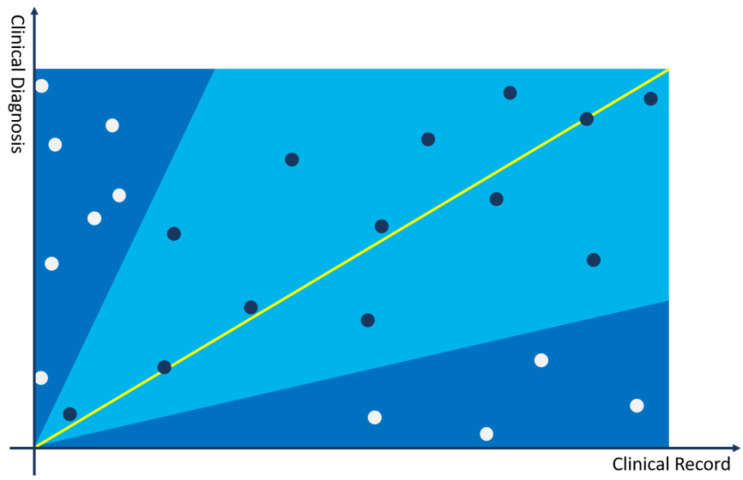
Cartesian coordinate system for calculating differential diagnosis.

**Figure 2 bioengineering-11-00029-f002:**
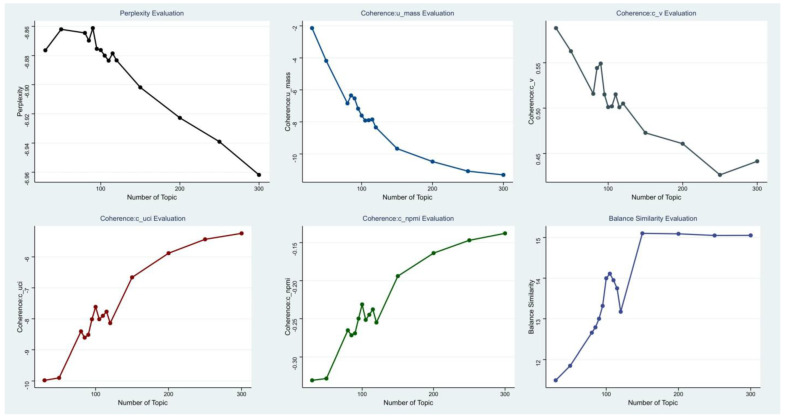
Various optimal topic evaluation curves.

**Figure 3 bioengineering-11-00029-f003:**
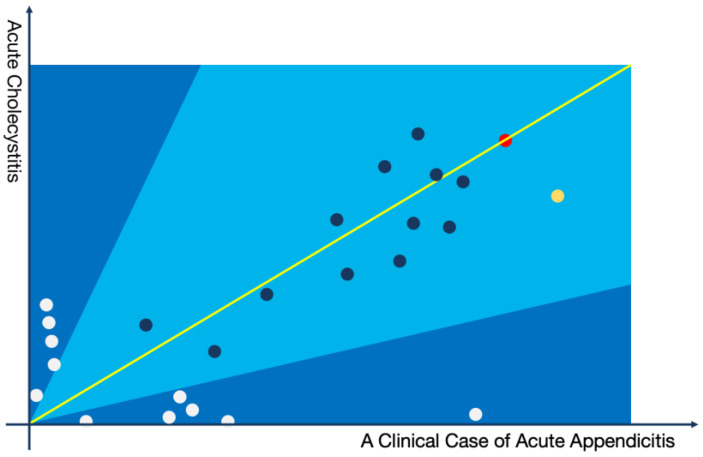
Graphical representation of a case of acute appendicitis with acute aholecystitis as a differential diagnosis.

**Table 1 bioengineering-11-00029-t001:** Model evaluation results under each topic number.

Number of Topic	Perplexity	Coherence:u_mass	Coherence:c_v	Coherence:c_uci	Coherence:c_npmi	Balance Similarity
30	−6.87642242	−2.139474262	0.588161857	−9.979548551	−0.330698141	11.4901489
50	−6.862041251	−4.184983578	0.562690422	−9.900777162	−0.328332816	11.84762616
80	−6.864519644	−6.840832494	0.515771135	−8.404860552	−0.265222598	12.65740147
85	−6.869757555	−6.346085892	0.544138848	−8.606396274	−0.27173602	12.78981896
90	−6.861197461	−6.530789839	0.549113873	−8.515818674	−0.269195885	13.00116799
95	−6.875413719	−7.172518292	0.515011299	−8.014896144	−0.249829264	13.31727989
100	−6.876245633	−7.603171697	0.501090594	−7.615493666	−0.231214409	13.99658663
105	−6.880070013	−7.914200582	0.501966873	−8.013731995	−0.251481734	14.10978418
110	−6.88351449	−7.894659788	0.515126344	−7.902379583	−0.244774822	13.94907584
115	−6.878557852	−7.847522271	0.500918626	−7.769326132	−0.237740792	13.74945148
120	−6.883310092	−8.340739443	0.505031677	−8.139187027	−0.254968395	13.17306891
150	−6.901836329	−9.669419994	0.472740124	−6.662310787	−0.194020836	15.10260948
200	−6.922738856	−10.47846721	0.460747388	−5.88178047	−0.163788813	15.09132694
250	−6.939098209	−11.078098	0.426396054	−5.433815865	−0.146954779	15.05041287
300	−6.961840108	−11.30937244	0.441389928	−5.241178775	−0.137830459	15.05255057

**Table 2 bioengineering-11-00029-t002:** Proposal methodology calculation results.

	Number of Hits/Total Differential Diagnosis Number	Recall
Top 5	3186/3645	0.916
	0.874	
Top 10	3397/3645	0.997
	0.932	

**Table 3 bioengineering-11-00029-t003:** Comparison of calculation results based on different numbers of topics.

	Recall (Number of Topic = 90)	Recall (Number of Topic = 150)	Recall (Number of Topic = 105)
Top 5	3186/3645	3051/3645	0.916
	0.874	0.837	
Top 10	3397/3645	3404/3645	0.997
	0.932	0.934	

**Table 4 bioengineering-11-00029-t004:** Comparison of the results between the proposal method and similarity calculation based on topic model.

	Recall (Cosine Similarity by Topic Model)	Recall (Proposal Method)
Top 5	2994/3645	0.916
	0.821	
Top 10	3259/3645	0.997
	0.894	

**Table 5 bioengineering-11-00029-t005:** Results of the Significance Difference Analysis between Training and Validation Datasets.

	Training Dataset (Amount of Training Data = 15,972)	Test Dataset (Amount of Testing Data = 1500)	p
Gender			0.183
Male	8349 (52.27%)	811 (54.07%)	
Female	7623 (47.73%)	689 (45.93%)	
Age	52 (17.97)	54 (23.89)	0.654
Marital Status			0.125
Single	4178 (26.16%)	406 (27.07%)	
Married	9473 (59.31%)	880 (58.67%)	
Widowed	360 (2.25%)	46 (3.06%)	
Divorced	1961 (12.28%)	168 (11.2%)	
Other	0 (0%)	(0%)	
Blood Type			0.997
A	4481 (28.06%)	417 (27.80%)	
B	3829 (23.97%)	360 (24.00%)	
O	6548 (41.00%)	618 (41.20%)	
AB	1114 (6.97%)	105 (7.00%)	
Rh			
Negative	8 (0.05%)	0 (0%)	1.000
Positive	15,964 (99.95%)	1500 (100%)	

**Table 6 bioengineering-11-00029-t006:** Results of the Differential Diagnosis Accuracy and Significance Differences in Different Population Groups.

Group	Age (Mean ± Std)	Test Dataset (Amount of Testing Data = 1500)	p	Proposal Method: Differential Diagnosis Recall (Top 5)
0–18 & Male	8.05 ± 5.66	197 (13.13%)	<0.001	0.899
0–18 & Female	7.73 ± 5.15	209 (13.93%)	<0.001	0.887
19–65 & Male	40.42 ± 13.19	441 (29.40%)	<0.001	0.921
19–65 & Female	40.17 ± 13.24	402 (26.80%)	<0.001	0.933
66+ & Male	73.85 ± 6.16	124 (8.27%)	<0.001	0.913
66+ & Female	74.64 ± 6.04	127 (8.47%)	<0.001	0.908

## Data Availability

Data cannot be shared due to privacy protection. The data studied in this paper are the medical records of real patients in the hospital, and due to the protection requirements of personal privacy, this part of the data cannot form a public dataset.

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
