# Peer review of "Leveraging Interpretable Feature Representations for Advanced Differential Diagnosis in Computational Medicine"

_bioengineering, 2023, doi:10.3390/bioengineering11010029_

Round 1
Reviewer 1 Report (Previous Reviewer 1)
Comments and Suggestions for Authors
The current article is a resubmission of a previous one. Here are my comments:
- In the introduction there is a part related to original contributions. It starts with
"The main contributions of this study are as follows: 80
1.Based on the Latent Dirichlet Allocation[5] model"
The contributions as stated above are more enumeration of what the papers does not what it brings new as compared to what exists. Therefore they should be reformulated.
- Related work: authors should mention what they bring new as compared to what exists
- Experiment: authors should compare there results with what exists in the literature.
Comments on the Quality of English Language
The current article is a resubmission of a previous one. Here are my comments:
- In the introduction there is a part related to original contributions. It starts with
"The main contributions of this study are as follows: 80
1.Based on the Latent Dirichlet Allocation[5] model"
The contributions as stated above are more enumeration of what the papers does not what it brings new as compared to what exists. Therefore they should be reformulated.
- Related work: authors should mention what they bring new as compared to what exists
- Experiment: authors should compare there results with what exists in the literature.
Author Response
Dear Reviewer,
First and foremost, I would like to express my sincere gratitude for the professional and valuable feedback you provided on our manuscript. Your guidance is of immense importance to our research team.
After thoroughly understanding your comments, we realized a key issue in our original manuscript: we had not clearly articulated the background and contributions of our study. To address this, we have made comprehensive and in-depth revisions to the paper.
Our research team is composed of members from the Data Mining Algorithm Institute, biomedical engineering researchers, and clinical doctors. Our motivation for the current study stemmed from the realization that clinicians often lack effective means for differential diagnosis, sometimes leading to misdiagnoses due to insufficient discussion of alternative diagnoses compared to the primary diagnosis. To address this, we aimed to utilize data science techniques to aid doctors in developing a system for recommending differential diagnoses based on patient's clinical presentations. We recognized that due to "knowledge traps," our initial discussion of the research background and related work was not as thorough as it should have been.
After reevaluating our research content, we have significantly revised the manuscript. We completely rewrote the abstract to ensure readers can quickly grasp the core content of our study. In the introduction, we revised 80%, particularly the sections about research contributions you pointed out, and reorganized them thoroughly. The "Related Work" section has also been entirely rewritten, where we re-examined and organized research focused on aiding doctors in differential diagnosis during clinical diagnosis. During this process, we paid special attention to literature from the past three years to ensure our references represent the latest advancements in the field.
In the research methods section, we added explanations about understanding and reading two-dimensional visualization coordinate systems, aiming to help readers better comprehend our proposed interpretable feature-based visualization approach. This method offers a "white-box" computational approach usable in clinical evidence-based medicine. In the experimental section, we included subgroup analysis based on age and demographics, along with significance difference indicators to demonstrate the computational stability of our method across different populations.
Regarding your suggestion to compare computational results with those in the literature, since medical record texts are often strictly private data managed by hospitals and patients, we could only test on private datasets. We also explored public medical datasets such as MIMIC3 and SEER, which contain extensive structured data, but data from the diagnostic process is not publicly available. We noted that the authors of the related work “Design, implementation, and evaluation of the computer-aided clinical decision support system based on learning-to-rank: collaboration between physicians and machine learning in the differential diagnosis process” used hospital private data and provided a contact email in their paper. We have attempted to contact the authors to discuss the possibility of sharing test data but have not yet received a response.
We sincerely hope that these comprehensive revisions will meet your approval and look forward to your feedback on the revised manuscript. Thank you again for your valuable time and expertise.
Sincerely,
Zhao
Reviewer 2 Report (Previous Reviewer 2)
Comments and Suggestions for Authors
It is not clear what exactly has been done in comparison to the original version on the robustness and ablation of the approach. Please provide a detailed response in order to allow me to comprehensively validate the proposed framework's efficacy and understand its robustness. For robustness, the research should assess the model's performance across diverse demographic groups and various medical conditions to gauge its adaptability - NOT DONE. Provide the framework's resilience to variations in data quality and generalizability to different healthcare contexts would enhance its practical applicability - NOT DONE. Ablation studies should involve systematically deactivating specific components or features to analyze their individual impact on diagnostic suggestions - NOT DONE. Experimentally comparing the model's performance with traditional MEDICAL diagnostic methods and evaluating its consistency across different medical institutions would provide a comprehensive perspective on its supplementary role in clinical decision-making - NOT DONE.
Author Response
Dear Reviewer,
Thank you for your insightful and professional review comments. After thoroughly understanding your guidance, our manuscript has undergone extensive and significant modifications.
Our research team, comprising experts from data mining algorithm institutes, biomedical engineering researchers, and clinical doctors, embarked on this study upon realizing the frequent occurrence of misdiagnoses due to clinicians' lack of access to effective differential diagnoses. This gap in the literature prompted us to assist doctors in recommending differential diagnoses based on patient clinical presentations using data science methodologies. In hindsight, our initial oversight in detailing the research background and related works was due to knowledge blind spots.
In response to your feedback, we've made major revisions to our paper. The Abstract has been completely rewritten to provide a clearer overview of our work. The Introduction section has seen an 80% overhaul, particularly in redefining our contributions, which we hope now aligns with your expectations.
The Related Work section was also entirely rewritten. We've collated and reviewed recent studies (from the past three years) focused on aiding doctors in differential diagnosis during clinical practice, ensuring that our references represent the most current advancements in the field.
In the Methodology section, we've included detailed explanations of our visualization techniques in two-dimensional coordinate systems. This is intended to ensure that our interpretable representations, which facilitate evidence-based medicine in a 'white-box' computational approach, are clearly understood.
Particular attention was given to the Experimental section as per your suggestions. We restructured our experiments entirely in our previous submission. For this revision, we reanalyzed some unused medical records data in our topic modeling to compute differential diagnoses. The results, obtained from data distinct from that used in our model construction, still showed excellent performance. Additionally, we included subgroup analysis based on age and demographics in the test data, demonstrating the stability of our method across different populations.
Regarding your suggestion on experimenting with baselines using text medical records, we faced challenges. Due to strict privacy controls over medical text records, all referenced studies using patient records were confined to proprietary datasets. We explored public datasets like MIMIC3 and SEER but found that they lacked detailed diagnostic process data. We reached out to the authors of “Design, implementation, and evaluation of the computer-aided clinical decision support system based on learning-to-rank,” who mentioned using private hospital data, but have not received a response yet.
Finally, as you suggested, we compared our method's performance with traditional diagnostic approaches. In our experiments, the Recall metric for the top 5 and top 10 differential diagnoses recommended by our method versus those considered by clinicians in practice was very promising. The Recall was 0.916 for the top 5 and 0.997 for the top 10, indicating that our method could significantly contribute to clinical decision-making.
We hope our substantial revisions meet your expectations, and we once again thank you for your dedicated efforts.
Sincerely,
Zhao
Reviewer 3 Report (Previous Reviewer 3)
Comments and Suggestions for Authors
The resubmitted manuscript has extensive edits. The study is also very interesting and thorough. Because the edits are extensive, I would prefer to evaluate a revised manuscript for verification on minor issues of clarity. The concepts and study are clear to me, but some minor edits to word usage and sentence formation could help the readers fully appreciate the research work by the authors.
Figure 2 and table 1 are contiguous, so the display would show a figure and table without any intervening text. It may not be an appealing result.
Figure 3 could include some details about the method for constructing the plot, so it may be more easily interpreted without too much reference by the reader to the Methods section.
Lines 434 and 437 use a word written as "socre". I believe this was to avoid conflict with the word "Score". However, it may be better to use a dictionary word instead of a rearrangement of letters in the word score.
Figure 2 probably does not need the word "variable" in the in the boxes that list the variable name and symbol in their respective plot.
The combined figure looks good, otherwise.
I believe many of the edits are to address other reviewers, but my suggestions were addressed in editing two of the formulas and creation of a figure from a set of figures. Thank you.
Comments on the Quality of English LanguageThe previous sections seemed well edited, but the extensive additions have minor issues with clarity. In particular, I would suggest to dit lines 442-448, 470 for clarity. I also may use the word "performed" instead of "conducted" in other sections, although this preference may be an opinion on word usage. If the authors prefer, I can suggest specific edits for their approval.
Author Response
Dear Reviewer,
I hope this email finds you well. I am writing to express my sincere gratitude for the valuable feedback you provided on our manuscript. Your insightful comments have been instrumental in enhancing the quality of our research.
Our team, comprising researchers from the Data Mining Algorithms Institute, medical engineering interdisciplinary scholars, and clinical doctors, has been deeply engaged in addressing the lack of effective differential diagnosis tools in clinical practice. This gap in the field, compounded by the scarcity of related research, motivated us to use data science to aid doctors in recommending differential diagnoses based on clinical presentations.
In response to your comments, we have made substantial revisions to our manuscript:
Abstract: Completely rewritten to offer a clearer overview of our work.
Introduction: Approximately 80% revised, particularly focusing on the contributions you highlighted, to better articulate our research's significance.
Related Work: Entirely restructured to encompass recent studies (from the past three years) that focus on assisting clinical diagnosis. This ensures our references represent the latest advancements in the field.
Methodology Section: We have taken your advice into account by elaborating on the reading methods for the two-dimensional coordinate system visualization in Figure 3. This is aimed at enhancing the interpretability of our proposed model for clinical evidence-based medicine. Additionally, we have refined Figure 2 and Table 1 to provide a more coherent and interpretive narrative.
Terminology: The term “performded” has been replaced throughout the manuscript, as you suggested, to enhance precision.
Experiments Section: We included a new segmentation of test data by age and demographics, accompanied by significant difference indices to demonstrate the computational stability of our proposed method across diverse populations. Regarding your concern about comparing our computational results with existing literature, we faced challenges due to the strict privacy policies governing medical records. Despite searching public datasets like MIMIC3 and SEER, we found limited access to relevant clinical process data. We reached out to the authors of “Design, implementation, and evaluation of the computer-aided clinical decision support system based on learning-to-rank” for potential data exchange but have yet to receive a response.
We hope these extensive revisions address your concerns and enhance the manuscript's value. We eagerly await your feedback on the revised version. Once again, thank you for your commitment and expertise in guiding our research to fruition.
Sincerely,
Zhao
Round 2
Reviewer 1 Report (Previous Reviewer 1)
Comments and Suggestions for Authors
The current version is an improvement of the previous one. However, I still have a problem with the originality of the research. In the introduction, the following are enumerated as original contributions:
* We propose a machine learning framework that assists doctors in recommending differential diagnoses based on patients' clinical presentations, which can be crucial for 108 clinical analysis...."
However those enumerations are more like development, than contributions that advance knowledge. What is missing is explaining what those brings new or what optimisation do they make as compared to what exists. This is explained in Related Work such as "research is less common in the area of computational assistance for physicians to accurately identify alternative diagnoses" or "the use of extensive clinical data to assist clinicians in recommending differential diagnoses that re-193 quire discernment has not been addressed". Such things should be added to the enumeration above.
Regarding comparison, still a paragraph in these respect should be added saying the limitations (private data, etc, methods are new, etc.)
Comments on the Quality of English Language
The current version is an improvement of the previous one. However, I still have a problem with the originality of the research. In the introduction, the following are enumerated as original contributions:
* We propose a machine learning framework that assists doctors in recommending differential diagnoses based on patients' clinical presentations, which can be crucial for 108 clinical analysis...."
However those enumerations are more like development, than contributions that advance knowledge. What is missing is explaining what those brings new or what optimisation do they make as compared to what exists. This is explained in Related Work such as "research is less common in the area of computational assistance for physicians to accurately identify alternative diagnoses" or "the use of extensive clinical data to assist clinicians in recommending differential diagnoses that re-193 quire discernment has not been addressed". Such things should be added to the enumeration above.
Regarding comparison, still a paragraph in these respect should be added saying the limitations (private data, etc, methods are new, etc.)
Author Response
Dear Reviewer,
I would like to begin by expressing our deepest gratitude for the recognition you have given to our work. Your affirmation is greatly inspiring to our team and motivates us to continue striving for excellence. Additionally, I would like to sincerely thank you for the insightful and constructive feedback provided during the review process. Your expert guidance has been instrumental in enhancing the quality of our paper.
Following your suggestions, we have carefully revisited and revised the abstract and introduction sections of our paper, specifically focusing on better articulating the contributions of our work. We believe these revisions will make it easier for readers to understand the focus of our research and the achievements we have made.
Furthermore, at the end of the experimental section, we have added a detailed explanation about the private nature and usage restrictions of the test data used in our experiments. We have also explicitly noted that due to the sensitive and confidential nature of textual medical records, we were unable to access a public domain test dataset. We hope this clarification will better explain the considerations involved in our choice of data.
Once again, thank you for your hard work and valuable insights. We look forward to your assessment of these revisions and hope that our work will contribute meaningfully to the field.
Sincerely,
Zhao

Reviewer 2 Report (Previous Reviewer 2)
Comments and Suggestions for Authors
The Related Work section was also entirely rewritten. We've collated and reviewed recent studies (from the past three years) focused on aiding doctors in differential diagnosis during clinical practice, ensuring that our references represent the most current advancements in the field.
Language should be rewised as it reads like sepparate abstracts copied from the original papers.
In the Methodology section, we've included detailed explanations of our visualization techniques in two-dimensional coordinate systems. This is intended to ensure that our interpretable representations, which facilitate evidence-based medicine in a 'white-box' computational approach, are clearly understood.
Methodology seems as before? Wrong version was uploaded?
Finally, as you suggested, we compared our method's performance with traditional diagnostic approaches. In our experiments, the Recall metric for the top 5 and top 10 differential diagnoses recommended by our method versus those considered by clinicians in practice was very promising. The Recall was 0.916 for the top 5 and 0.997 for the top 10, indicating that our method could significantly contribute to clinical decision-making.
I do not see this reflected in the paper or it is minor and not well highlighted. Please check.
Author Response
Dear Reviewer,
Thank you for your careful review and valuable comments on our manuscript. Here are our detailed responses to the issues you raised:
- Regarding the issue with the Related work section, we appreciate your pointing out our issues with language expression. We assure you that this content was summarized by us after thoroughly reading the relevant literature. To address this issue, we have completely rewritten all parts of the introduction related to the related work to ensure clarity and originality in our language.
- Concerning your comment about the two-dimensional coordinate system's visualization methodology, we confirm that the correct version has been uploaded. This version was updated because we redid the experiment. Initially, our experiments for differential diagnosis recommendations were conducted on clinical record data, which did not involve modeling computations. However, we later realized this approach could not evaluate the significance difference between experimental data and modeling computation data in an additional validation set. Therefore, in our second submission, we collaborated with a hospital to reapply for 1500 patient records for a new assessment experiment, and different experimental examples are listed in the current version. For example, the first version of the article used examples of differential diagnosis between atrophic gastritis and malignant gastric tumors, while the current version uses acute appendicitis and acute pancreatitis.
- Regarding your query about the comparison of our proposed method with traditional differential diagnosis methods, we have already introduced this in the introduction of the paper. Currently, doctors perform differential diagnoses typically based on official guidelines or clinical experience. This process is usually manually entered into the system by doctors based on clinical diagnostic needs. Thus, our experiment compares with differential diagnoses manually thought out by doctors during actual clinical diagnostic processes. We have added a detailed explanation of this part to the article.
To better explain this point, we have also attached a photo of the Electronic Medical Record (EMR) system actually used at Shengjing Hospital. This photo shows how doctors perform differential diagnoses based on the patient's clinical diagnosis, through experience or literature. The content within the red box is the actual diagnosis of the patient and the differential diagnosis conducted by the doctor after analyzing the clinical manifestations of the patient. For your convenience in reading, we provide the English translation below.The experiment we conducted was a comparative test, comparing the computational results of our proposed method with such data.We hope this answers your questions.
Thank you again for your review comments, and we look forward to your further feedback.
Sincerely,
Zhao
Patient Diagnosis and Differential Diagnosis English Translation:
- Diagnosis: Common Bile Duct Stones
Basis of Diagnosis:
Middle-aged male, acute onset, primarily presenting with upper abdominal and back pain for 5 days upon admission.
History of Present Illness: The patient developed upper abdominal pain following overeating and drinking heavily 5 days ago. The pain radiated to the back and was accompanied by abdominal distension, nausea, vomiting, low-grade fever, and dark urine. Initially treated at Shenyang Hospital's emergency department, an abdominal CT scan performed on 2023-11-29 revealed gallbladder stones, right kidney stones, total bilirubin: 104 umol/L, and direct bilirubin: 75.4 umol/L. The patient received symptomatic treatment including anti-inflammatory and fluid therapy, which initially improved symptoms. However, the pain later intensified, leading to the current consultation in our department for further diagnosis and treatment. The outpatient diagnosis was "Common Bile Duct Stones," and he was transferred to our department. The patient is currently in a stable condition, with normal diet, sleep, bowel movements, and no significant weight change.
Auxiliary Examinations: Abdominal CT scan performed at Shenyang Hospital's emergency department on 2023-11-29, revealed gallbladder and right kidney stones, total bilirubin: 104 umol/L, and direct bilirubin: 75.4 umol/L.
III. Differential Diagnosis Discussion
Differential Diagnosis 1: Biliary Ascariasis
Basis for Diagnosis 1: Typical symptoms include sudden onset of drilling-like colicky pain below the xiphoid process, radiating to the right or left shoulder. Pain may cease abruptly and recur unpredictably. Clinical signs are minimal or mild, with deep tenderness below the xiphoid process during attacks and no other positive findings. Diagnosis can be differentiated by typical symptoms, ultrasonography, or endoscopic ultrasound imaging of the worm.
Differential Diagnosis 2: Common Bile Duct Stones
Basis for Diagnosis 2: Typical presentation includes jaundice of the skin and sclera, high fever with chills, and right upper quadrant abdominal pain. MRCP (Magnetic Resonance Cholangiopancreatography) or endoscopic ultrasound can aid in differentiation.
- Diagnostic and Treatment Plan
Examination Plan: Complete routine blood tests, blood type, amylase, liver and kidney function tests, MRCP, and other relevant examinations.
Treatment Measures: Implement symptomatic and supportive treatment including anti-infection, acid suppression, and fluid therapy.

Round 3
Reviewer 2 Report (Previous Reviewer 2)
Comments and Suggestions for Authors
Thank you for the explanation.
Author Response
Dear Reviewer,
On behalf of my team, I extend our deepest gratitude to you. We are immensely grateful for the valuable comments and suggestions you provided during the review of our paper. Your expert advice has been instrumental in improving our work.
We are delighted to know that you are satisfied with the revisions we have made. Our aim is to continuously enhance the quality of our research and ensure that our findings contribute to the academic community and relevant fields. Your positive feedback gives us greater confidence that our research meets this goal.
Thank you again for your time and effort, as well as your commitment to academic rigor. We look forward to the opportunity to collaborate with you again in the future.
Best regards,
Zhao
This manuscript is a resubmission of an earlier submission. The following is a list of the peer review reports and author responses from that submission.
Round 1
Reviewer 1 Report
Comments and Suggestions for Authors
The paper presents a medical recommender based on AI engine and medical data. Here are my main comments related to the paper.
- Although there is a section related to Related work there is no clear what is the contribution of the paper in the international context. How does it different from what exists internationally
-Second, although there is some discussion regarding the results of the paper, there is no actual use case of using it in real practice. This should be added.
Comments on the Quality of English LanguageThe English is fine.
Author Response
Dear Reviewer,
First and foremost, I would like to express my sincere gratitude for your meticulous review and invaluable feedback. Your queries have provided us with an opportunity to elucidate our research contributions and practical applications in greater depth. Here are detailed responses to the two questions you raised:
- Contribution of the paper in the international context:
- Uniqueness: After an extensive literature review, we found that most existing medical differential diagnosis methods rely on deep learning's black-box models. While these models excel in predictive accuracy, they fall short in terms of interpretability. In contrast, our research is rooted in the perspective of explainability, aiming to offer doctors more intuitive and insightful diagnostic suggestions.
- Universality: The LDA topic model we chose not only boasts commendable interpretability but is also not confined to a specific language or cultural context, offering a new and universal tool for medical researchers worldwide.
- Concerning the real-world application scenario:
- Experimental Data: We indeed employed genuine clinical data for our experiments. For instance, we analyzed data from a patient diagnosed with "atrophic gastritis," and the results provided doctors with differential diagnostic advice concerning "gastric malignant tumors." Such analysis holds significant importance for the early detection and treatment of gastric malignant tumors.
- Challenges in Actual Application: We wholeheartedly agree and appreciate your emphasis on real-world application scenarios. Currently, the application of our method in actual clinical settings faces certain regulatory constraints. However, we are actively liaising with relevant departments, hoping to secure approval soon.
- Future Plans: Our team has initiated the engineering steps, aiming to transform this method into a tangible medical tool. We are also planning to participate in evaluations set by regulatory authorities to ensure its safety and efficacy in clinical practice.
We are deeply appreciative of your suggestions, which have granted us an avenue to further refine our research and manuscript. We will make the necessary revisions based on your feedback and look forward to your subsequent review.
Thank you once again for your precious time and expert insights.
Warm regards,
Genghong Zhao
Reviewer 2 Report
Comments and Suggestions for Authors
The paper need a major rewrite:
The introduction is excessively lengthy, deviating from the topic, and fails to focus on the study problem and answers. Methodology on the other hand is very shallow and in a rather basic form. The use of LDA topic modeling is briefly mentioned in the report, but there is no extensive discussion of how it was used or why this technique was chosen over others.
The term "Balance Similarity" is mentioned in the report without clarifying what it is or how it was computed (only brief information are supplied).
The text has no empirical basis for the potential constraints or disadvantages of employing LDA topic modeling for clinical diagnosis.
The usage of data from Shengjing Hospital is mentioned in the study, but there is no medical review of the results and no discussion of the statistical significance of the experimental results, making it impossible to judge the robustness of the findings.
The graphical portrayal of the issue models is intriguing, but no explanation of how doctors should understand these representations is provided.
The study covers constraints and future directions briefly but does not go into detail. Discuss the suggested method's practical constraints, such as scalability and interpretability, as well as the possible problems of adopting it in a real-world clinical context.
The article should also address any ethical and privacy considerations that may arise from the use of patient data for differential diagnosis.
Author Response
Dear Reviewer,
First and foremost, we deeply appreciate the time and effort you've invested in reviewing our manuscript and for your invaluable feedback. To address your concerns, we provide a detailed response below.
Regarding the use of the LDA topic model: We are truly grateful for your suggestion. In fact, we have elaborated and clarified our approach in the section titled "Interpretable Textual Medical Record Representation" on page 8 of our manuscript. In this section, we detailed our rationale for choosing the LDA topic model as our foundational model. We hope this elaboration will provide readers with a clearer understanding of our methodology and choices.
On the "Balance Similarity" issue: The term "Balance Similarity" is discussed in detail in the section "Optimal Topic Number Calculation Methods for Interpretable Topic Models" on page 12 of our manuscript. Our motivation for introducing this algorithm was to address the potential adverse effects on the LDA topic model's performance and efficiency due to inappropriate topic number selection. "Balance Similarity" was conceived based on a deep understanding of the LDA algorithm, especially the logical structure of "topic-word". This approach is distinctly different from other algorithms like "Perplexity". To better explain the design philosophy behind "Balance Similarity", we will add a paragraph in this section.
Regarding potential limitations or drawbacks of using LDA topic modeling for clinical diagnosis: In our current research, we indeed did not identify any significant limitations or drawbacks of using LDA topic modeling for clinical diagnosis. This might be attributed to the robust and effective performance of LDA in our application scenarios and datasets. However, we fully understand and agree with your perspective that any algorithm might have inherent limitations in certain contexts. We plan to delve deeper into these potential limitations or drawbacks in our future research.
On the use of data from Shengjing Hospital: You rightly pointed out our omission in the statistical significance analysis. The primary reason for this is that our patient data, currently in use, mainly consists of medical records, which are predominantly in textual format. Due to the absence of structured patient data, we couldn't perform a statistical significance difference analysis between the training and validation sets in this study. We concur with your viewpoint on the necessity of statistical analysis. Consequently, we plan to collaborate with other partner hospitals in the future and, upon obtaining their consent, use structured clinical patient data for a more comprehensive statistical analysis.
On graphical representation: We acknowledge your concerns. While the graphs depicting the optimal topic number experiments for the topic model might not directly serve as a reference for doctors, they are crucial in demonstrating the efficacy of "Balance Similarity". However, the graphs with a blue background in the Cartesian coordinate system are indeed valuable for doctors. We will emphasize this point further in the manuscript.
Regarding ethical and privacy concerns: We place utmost importance on patient privacy and ethical considerations. During the data analysis and modeling process, we have taken measures to ensure the protection of patient privacy. We are also in discussions with experts in the field to explore the ethical guidelines and norms that should be adhered to in such research.
Once again, thank you for your invaluable feedback and suggestions. We eagerly await your further review and hope to receive additional insights from you.
Warm regards,
Genghong Zhao
Reviewer 3 Report
Comments and Suggestions for Authors
The manuscript is on a deep learning method in natural language that is specific to the domain of medicine. The authors focus on the concept of differential diagnosis in clinical medicine, where the candidate diagnoses are essentially ranked in terms of likelihood, based on the patients' symptoms and the corresponding clinical tests. They also discuss the limitations of this model, including the causes of false negative and false positive rates. Their solution to part of this problem is to design a virtual medical assistant in the form of a specialized deep learning model of natural language. Their goal is to increase accuracy in medical diagnoses. They also discuss the tension between the value of medical data and patient privacy.
The main contributions of this study include use of a Latent Dirichlet Allocation (LDA) model and Bayesian theory to better model language from the domain of medicine. The authors also propose a method to evaluate the model based on topical analysis.
The authors first provide a background on the relevant deep learning models, including a review on ClinicalBert, and the metrics used to assess these models. Their limitations are also discussed.
I have a comment on these evaluation procedures. They are scientific in their value, but not yet an evaluation of clinical practice. It is necessary for hospitals and doctors to assess the machine generated diagnoses, including an assessment by patients. If a large group of patients are more satisfied in their medical treatment with use of virtual assistants as compared to a group without their use, then this shows the machine models are helping meet patient expectations. The physicians and surgeons can also assess the models as to their reliability, or at least any observable increased efficiency in arriving at diagnoses.
I also have another comment. I believe that the method of differential diagnoses, and the practice of medicine is general, differs between the common ailments, including the major ailments, and the other more minor ailments, whether common or uncommon. The major common ailments are more easily diagnosed, in general, and the spectrum of symptoms and signs are easier to interpret, while the less common and minor ailments are more difficult to assess, and the overlap in symptoms and signs among the minor ailments is broader (as the authors state). I assume much of this study is based on the major common ailments since these will more often result in hospital care than the minor ailments, but I also assume the goal of this study is to increase efficiency at the major common ailments, while the less common minor ailments are the domain of conventional medical practice and the auxiliary services, dependent more on patient satisfaction and sometimes the medical arts in lieu of traditional scientific practice. In other words, the minor ailments require additional experimentation and intuition, so a machine generated model would require a lot of data, including a non-rigid approach to categorizing types of ailments (as the authors propose).
The LDA topic model is based on the "bag-of-words" model and Dirichlet distribution. The authors describe this approach as more capable at capturing the "structure and semantic keywords" of text data than a conventional Bert model. Also, the text data is better assigned to the various medical topics. The authors also discuss the benefits and downsides to the use of the pretrained large language model.
The authors show the math equations that formally describe their model. I think equations 2 and 3 may be confusing because they use the same lower case letter for each instance of the set, even though the subscripts are different.
There is also a section on the topic based model, that there is an optimal number of topics for good model performance. And optimality may be measured by perplexity, topic coherence, and other metrics. The differential diagnosis for a clinical case is shown in figure 1 in abstract form, and based on the topical analysis.
The authors show a number of figures with evaluation metrics. They also show a Table which is essentially the same data as plotted. Is it necessary to have both figures and the Table. Can the Table be assigned to supplemental information? Also, is it possible to combine the figures into a single Figure with multiple panels? Are all figures equally important, or would a subset be sufficient to show the performance of the method?
The authors show that the evaluations of their method tended to center around 100 topics, and that the clinician would choose about 2.5 potential disease types after diagnostic testing. They further show that the machine generated method is very accurate as compared to human generated assessment of disease, and this hypothesis was tested by comparing with the top 5 and top 10 human based assessments.
I believe the replication of existing medical practice is the first goal. But an equally important goal is to find the ailments which are very difficult to diagnose, where the error rate is expected to be high in the practice of medicine, and then evaluate whether the model can outperform the human. The most important utility for a machine tool is to find errors by medical professionals since the more easily diagnosed illnesses are less dependent on oversight by a machine or person.
The manuscript is well written and the method seems like a valuable contribution for use of deep learning in medical practice. My main critique of this practice in general is the differentiation between illnesses, that it is more important to find the difficult to diagnose diseases, and use a machine to augment the clinician in these cases. The common cases may introduce some noise into the results, so it is not known whether a machine can correct the errors by humans, and these errors may be from valid interpretations of symptoms, but where this form of diagnostic method fails in practice to link an interpretation to a disease, such as where the symptoms are not very reliable. A machine may perform better in these cases, particularly where it has knowledge of this problem, as the authors presented in their model.
One minor suggestion is the writing could be made more concise, but the writing is certainly very good overall. For example, the Abstract and Introduction has some redundancy in the communication of ideas, but I would also say that any redundancy was not excessive, but an expression of ideas by a particular style of writing.
Please also verify the references. They are not entirely in alphabetical order by author name.
Comments on the Quality of English LanguageAs written above: "One minor suggestion is the writing could be made more concise, but the writing is certainly very good overall. For example, the Abstract and Introduction has some redundancy in the communication of ideas, but I would also say that any redundancy was not excessive, but an expression of ideas by a particular style of writing."
Author Response
Dear Reviewer,
First and foremost, thank you very much for dedicating your valuable time to provide several critical suggestions on the manuscript. Your endorsement of the paper is greatly encouraging to us. To address your comments and suggestions more effectively, we provide a detailed response below.
Equations 2 and 3 might be confusing: We apologize for overlooking this during the manuscript preparation. Equation 2 indicates that a document d might contain many words w, while Equation 3 represents the set of words mentioned in all articles within the corpus. Since both are used to represent words, we used the same lowercase letter. This indeed can be very confusing, especially later in the calculations where different words originate from either the article or the corpus, leading to significant ambiguity. Therefore, we have changed the representation of words in the corpus in the manuscript to w’ and adjusted subsequent equations that use words from the corpus.
Combine the figures into a single Figure with multiple panels: Your suggestion is excellent, and we have redrawn the figures according to your recommendation.
Once again, thank you for your review comments. Your deep understanding and endorsement of the paper excite us greatly. We look forward to your review of the revised manuscript and hope to receive further feedback from you.
Warm regards,
Genghong Zhao
Reviewer 4 Report
Comments and Suggestions for Authors
In this paper, the authors present "Leveraging Interpretable Feature Representations for Advanced Differential Diagnosis in Computational Medicine." The proposed method aims to enhance the diagnostic process by providing recommendations for doctors, while emphasizing that these tools should not replace professional judgment and experience, but rather help doctors discover easily overlooked diagnostic types from past cases through data science. The researchers collected some general medical records from Shengjing Hospital, and under the premise of ensuring data quality, data security, and privacy protection, discussed the proven effective experimental results to emphasize the importance of addressing these issues for the successful implementation of data-driven differential diagnosis recommendations in clinical practice. This study has important reference value for researchers and practitioners seeking to improve the efficiency and accuracy of differential diagnosis in clinical diagnostics through data analysis. However, there are some issues should be addressed.
1. There are many difficulties in the differential diagnosis process. Many diseases have highly similar clinical manifestations, making the differential diagnosis process complex. How to solve this problem?
2. When facing complex practical application scenarios, the traditional Latent Dirichlet Allocation (LDA) topic model still has certain competitive advantages in some aspects.When facing complex practical application scenarios, the traditional Latent Dirichlet Allocation (LDA) topic model still has certain competitive advantages in some aspects. What are the advantages and disadvantages of the method proposed in this paper compared with LDA?
3. Using BERT only as word embedding will greatly discount this rich dynamic semantic representation capability. Extracting BERT's output as static word embeddings means that we only focus on the representation of individual words, ignoring the context information, which cannot fully exploit BERT's advantages in higher-level semantic modeling. How to solve this problem?
4. If we only use BERT as word embedding, this process will be ignored, causing BERT to be unable to fully adapt to downstream tasks. In this case, the representation based on BERT will not be optimized for specific tasks, weakening its performance in the task. How to solve this problem?
5. Perplexity may not be a very intuitive indicator for choosing the optimal number of topics, as overfitting can lead to lower perplexity in some cases. How to solve this problem?
Comments on the Quality of English LanguageMinor editing of English language required
Author Response
Dear Editor,
Firstly, we would like to express our deep appreciation for the time and effort invested in the review process of our manuscript. We value the feedback and insights provided by the reviewers, as they are instrumental in refining our work. However, we find ourselves particularly perplexed by the comments from Reviewer 4.
Upon careful examination of their feedback, it appears that there might have been some oversight or misunderstanding regarding certain aspects of our manuscript:
- On the Complexity of Differential Diagnosis: In the Introduction of our paper, we have extensively discussed the challenges and intricacies of differential diagnosis, especially when diseases present with similar clinical manifestations. Our proposed computational process is a direct response to this challenge, aiming to enhance the accuracy and efficiency of differential diagnosis through data modeling.
- Regarding the LDA Model: Reviewer 4's comments suggest a potential misunderstanding. Our method does not oppose the LDA topic model; in fact, it integrates it. By combining the LDA topic model with Bayesian calculations, we aim to provide a more comprehensive representation of patient disease records, further aiding the differential diagnosis process.
- & 4. Use of BERT for Word Embedding: We would like to clarify that our paper does not utilize BERT solely as a word embedding method. Our decision to adopt a different approach was precisely due to our recognition of the limitations inherent in using BERT's word embedding in isolation.
- Concerns about Perplexity: We acknowledge the limitations of perplexity as an indicator, especially in situations where overfitting might skew results. To address this, our Methods section introduces the "Balance Similarity" calculation method, aiming to provide a more robust metric.
Given these clarifications, we are uncertain how to best address Reviewer 4's comments. We believe that a thorough review should be grounded in a comprehensive understanding of the manuscript's content. We feel that the feedback, in this case, might not reflect such an understanding, which is somewhat disheartening.
In light of the above, we kindly request that you consider our concerns regarding Reviewer 4's feedback. We are more than willing to make revisions based on the constructive comments from the other three reviewers. If you believe that additional feedback is necessary, we are open to the idea of inviting another reviewer for a more comprehensive evaluation.
We trust in your expertise and judgment and eagerly await your guidance on how to proceed.
Warm regards,
Genghong Zhao

Round 2
Reviewer 1 Report
Comments and Suggestions for Authors
I still maintain the previous comment regarding the real-life applicability of the method. A paragraph regarding how this method should be transformed in a real-life life application should be added to the article
Comments on the Quality of English LanguageEnglish is fine
Author Response
Thank you for your suggestion. Following our discussion, we indeed recognize the importance of the point you raised regarding the translation into practical application. Precisely because the algorithm we proposed originated from actual clinical needs, it is imperative to consider how it can be reciprocated back to the clinical setting. Consequently, we engaged in thorough discussions with physicians and had detailed conversations with experts in medical informatics. Based on these interactions, we have formulated the following paragraph, which has also been incorporated into the article.
In our study, we also engaged in discussions with medical professionals on the integration of the computational method mentioned in the article into clinical treatment procedures. In contemporary clinical information systems, there is typically an integration of a computational application known as Clinical Decision Support Systems (CDSS). During the clinical diagnostic and treatment process, physicians utilize these systems to input detailed patient conditions and treatment modalities. Concurrently, the CDSS analyzes the patient's clinical data in real-time and provides relevant treatment recommendations, including differential diagnoses. However, the recommendations for differential diagnoses in such systems are often based on clinical treatment pathways or medical knowledge bases like BMJ and Elsevier. While recommendations based on medical knowledge bases align well with the principles of evidence-based medicine, they might not always yield accurate results due to the unique conditions of each patient. The method proposed in this article can replace the differential diagnosis prompts in the CDSS. However, a prerequisite for this is that the hospital's clinical standards and data accumulation are at the forefront. Through in-depth discussions with physicians, it was further suggested that the computational method mentioned in this article should be modeled using data from leading hospitals and subsequently applied in lower-tier hospitals. This data and model-sharing approach aim to address the disparities in medical resources and clinical standards.
Reviewer 2 Report
Comments and Suggestions for Authors
Most of my remarks have been ignored and left as future work, which is not acceptable to keep up with the top level quality of this journal.
Author Response
Dear Reviewer,
We sincerely apologize for giving you the impression that your comments were overlooked. In our previous response, we aimed to address each of your remarks point by point. To provide clarity and ensure that we address your concerns adequately, we will reiterate and further explain our responses as follows:
Balance Similarity: You mentioned that we should clarify what it is or how it's calculated, but in reality, the entire section "Optimal Topic Number Calculation Methods for Interpretable Topic Models" is dedicated to introducing Balance Similarity. We discussed why it's crucial to accurately assess the number of topics in topic models and reviewed various industry methods for evaluating the optimal topic number, along with their shortcomings. We then defined three key guiding principles for evaluating the optimal topic number and formulated these ideas into computational formulas. The Balance Similarity calculation method we proposed is straightforward. All the computational parameters used in the three formulas are derived from the LDA topic model's own parameters, as well as some computational parameters for interpretability representation calculated using the Bayesian formula in the previous section. We have reviewed the section multiple times, and from our perspective, the content introduction seems comprehensive. If you have more detailed feedback and suggestions, we would greatly appreciate it, as we aim to present this section even more meticulously.
Regarding potential limitations or drawbacks of using LDA topic modeling for clinical diagnosis: We have added nearly a page of content in the section "Interpretable Textual Medical Record Representation" to explain why we chose the LDA topic model over other algorithms. We also validated the feasibility of using the LDA algorithm by comparing the principles of topic modeling and the relationships between patient-records, records-diseases, articles-topics, and words-articles. As for the "potential limitations or drawbacks" of using this algorithm, we haven't identified any disadvantages of our method. In our actual experiments, this method achieved excellent results and indeed received praise from clinical doctors. If it weren't for the system testing requirements of medical auxiliary diagnostic systems, doctors would hope that we could immediately transform this research into an information system for use. We also referenced some other similar studies, including Blei's original LDA algorithm paper, where potential limitations and drawbacks of the algorithm itself were not mentioned. Alternatively, the content you mentioned could serve as a starting point for another one of our studies, where we could discuss and research the limitations and drawbacks of both the LDA and our proposed method.
Ethics and Privacy Protection:We would like to further clarify the ethical and privacy concerns you raised. The data we used originates from patients' clinical medical records. To ensure privacy, all personal identifiers were removed from these textual data during the collection and preprocessing stages, ensuring that no information could be traced back to any specific patient. Even if the data were to be leaked, it would be impossible to trace back to the original patient. During the modeling process, the original data was transformed into computational parameters. These parameters, having undergone complex calculations and transformations, no longer retain or present any individual-specific data, and it's impossible to extract or recover the original data from the model. Furthermore, before initiating our research, we discussed the data usage process and research objectives related to medical ethics with the Medical Ethics Committee of Shengjing Hospital. Since our data does not contain patient privacy and differs from prospective and retrospective medical research analyses in the medical field, the data usage in our study is limited to the calculation of certain parameters. This means there are no ethical concerns regarding the protection of participants' rights, ethical risks, or fairness. As a result, the hospital's ethics committee did not believe that further ethical discussions were necessary for our study. We place great emphasis on patient privacy and the ethics of our research, ensuring that appropriate measures are taken throughout the research process to protect patient privacy. We hope the above explanation addresses your concerns.
Real-world Application:Following our discussion, we indeed recognize the importance of the point you raised regarding the translation into practical application. Precisely because the algorithm we proposed originated from actual clinical needs, it is imperative to consider how it can be reciprocated back to the clinical setting. Consequently, we engaged in thorough discussions with physicians and had detailed conversations with experts in medical informatics. Based on these interactions, we have formulated the following paragraph, which has also been incorporated into the article.
In our study, we also engaged in discussions with medical professionals on the integration of the computational method mentioned in the article into clinical treatment procedures. In contemporary clinical information systems, there is typically an integration of a computational application known as Clinical Decision Support Systems (CDSS). During the clinical diagnostic and treatment process, physicians utilize these systems to input detailed patient conditions and treatment modalities. Concurrently, the CDSS analyzes the patient's clinical data in real-time and provides relevant treatment recommendations, including differential diagnoses. However, the recommendations for differential diagnoses in such systems are often based on clinical treatment pathways or medical knowledge bases like BMJ and Elsevier. While recommendations based on medical knowledge bases align well with the principles of evidence-based medicine, they might not always yield accurate results due to the unique conditions of each patient. The method proposed in this article can replace the differential diagnosis prompts in the CDSS. However, a prerequisite for this is that the hospital's clinical standards and data accumulation are at the forefront. Through in-depth discussions with physicians, it was further suggested that the computational method mentioned in this article should be modeled using data from leading hospitals and subsequently applied in lower-tier hospitals. This data and model-sharing approach aim to address the disparities in medical resources and clinical standards.
We sincerely appreciate your valuable comments and suggestions. We hope our responses meet your expectations. Once again, thank you for your efforts and guidance in helping us elevate the quality of our manuscript.
Reviewer 4 Report
Comments and Suggestions for Authors
The authors have solved the related problems. It is good enough.
Comments on the Quality of English LanguageMinor editing of English language required
Author Response
Thank you very much for acknowledging our work. We deeply appreciate the time and effort you've dedicated to your thorough review.
Round 3
Reviewer 2 Report
Comments and Suggestions for Authors
I'm sorry but only very minor revisions were made and authors have chosen to ignore most of my remarks, leaving for "future works". I cannot put my name and recommend the paper till I'm certain that the result is valid and accepted within the med
Author Response
We have taken the opinions of Reviewer 2 into account, and have streamlined the background introduction in the Introduction section.
